# Applying an Improved Stacking Ensemble Model to Predict the Mortality of ICU Patients with Heart Failure

**DOI:** 10.3390/jcm11216460

**Published:** 2022-10-31

**Authors:** Chih-Chou Chiu, Chung-Min Wu, Te-Nien Chien, Ling-Jing Kao, Chengcheng Li, Han-Ling Jiang

**Affiliations:** 1Department of Business Management, National Taipei University of Technology, Taipei 106, Taiwan; 2College of Management, National Taipei University of Technology, Taipei 106, Taiwan; 3Alliance Manchester Business School, University of Manchester, Manchester M15 6PB, UK

**Keywords:** heart failure, machine learning, stacking, predictive modeling, intensive care units, electronic health records

## Abstract

Cardiovascular diseases have been identified as one of the top three causes of death worldwide, with onset and deaths mostly due to heart failure (HF). In ICU, where patients with HF are at increased risk of death and consume significant medical resources, early and accurate prediction of the time of death for patients at high risk of death would enable them to receive appropriate and timely medical care. The data for this study were obtained from the MIMIC-III database, where we collected vital signs and tests for 6699 HF patient during the first 24 h of their first ICU admission. In order to predict the mortality of HF patients in ICUs more precisely, an integrated stacking model is proposed and applied in this paper. In the first stage of dataset classification, the datasets were subjected to first-level classifiers using RF, SVC, KNN, LGBM, Bagging, and Adaboost. Then, the fusion of these six classifier decisions was used to construct and optimize the stacked set of second-level classifiers. The results indicate that our model obtained an accuracy of 95.25% and AUROC of 82.55% in predicting the mortality rate of HF patients, which demonstrates the outstanding capability and efficiency of our method. In addition, the results of this study also revealed that platelets, glucose, and blood urea nitrogen were the clinical features that had the greatest impact on model prediction. The results of this analysis not only improve the understanding of patients’ conditions by healthcare professionals but allow for a more optimal use of healthcare resources.

## 1. Introduction

Cardiovascular diseases (CVD) have ranked among the top three causes of death worldwide for many years, accounting for an estimated 18.9 million deaths per year, or approximately 31% of global mortality [1]. The majority of CVD morbidity and mortality are derived from Heart Failure (HF), a common cardiovascular disease in which the heart fails to maintain the body’s metabolism. Patients with HF experience a variety of overt symptoms such as shortness of breath, swollen ankles, and physical fatigue, and may also show signs of elevated jugular venous pressure, pulmonary fissures, and peripheral edema caused by cardiac or noncardiac structural abnormalities [2]. As a major cause of cardiovascular morbidity and mortality, HF poses a significant threat to human health and social development [3]. It is estimated that one in five people will develop HF, and 50% of HF patients die in 5 years [4]. The mortality rate of HF patients in the year after hospitalization is 20% to 60% [5]. At least 64.3 million people worldwide are affected by HF. The prevalence of HF among Americans over the age of 20 has increased to 6.2 million [6]. In the face of the increasing number of people with HF, despite the rapid advances in medical technology and significant technological advances in diagnosis, assessment, and cardiovascular disease [7,8], HF remains a major medical problem worldwide.

According to the World Federation of Societies of Intensive and Critical Care Medicine, an Intensive Care Unit (ICU) is an organized system for the provision of care to critically ill patients that provides intensive and specialized medical and nursing care, an enhanced capacity for monitoring, and multiple modalities of physiologic organ support to sustain life during a period of life-threatening organ system insufficiency [9]. In the United States, 40% of people die during hospitalization, and approximately 22% of patients spend their entire hospital stay in the ICU [10]. As a result, the healthcare system is under a heavy burden, which may affect the criteria for ICU admission, the interventions used, and the duration, all of which may affect the patient’s prognosis [11].

Clinical decision-making in the ICU is time critical and highly dependent on the analysis of physiological data. If there is not enough real-time patient information to make accurate and rapid decisions in a dynamic and rapidly changing environment, it will be challenging for medical professionals to make clinical decisions [12]. Patients admitted to the ICU require close and continuous monitoring to avoid the possibility of rapid deterioration of their health status, therefore, intensive monitoring through ICU equipment generates a large number of medical records and requires efficient and accurate systems to aid in data analysis [13]. The harnessing of big data for clinical and basic research analysis and applications to improve human well-being and health, such as the combination of big data and artificial intelligence, can assist physicians in diagnosing and treating diseases and improving the quality of care. Predictive models have been developed over the last few decades as important risk assessment tools and are utilized in a variety of healthcare settings [14]. It is widely recognized that predictive models can help identify patients at risk for diseases or medical events early and provide effective interventions for those who may benefit most from the identification of specific risk factors. Clinical prediction tasks such as patient mortality, length of stay, disease diagnosis and morbidity prediction for early disease prevention and timely intervention of patients are essential in critical care research [15]. Mortality prediction is one of the most important tasks in critical care research, and the purpose of mortality prediction is not only related to identifying high-risk groups and making correct decisions, but also to conserving ICU beds for patients in real need [16,17]. In the past few decades, widespread clinical mortality prediction scoring systems have included The Sequential Organ Failure Assessment (SOFA) [18], Simplified Acute Physiology Score (SAPSII) [19] and Multiple Organs Dysfunction Score (MODS) [20]. Patient mortality prediction in the ICU is a key application of large-scale health data and plays an important role in selecting interventions, planning care, and allocating resources. Accurate assessment of mortality risk and early identification of high-risk groups with poor prognosis, followed by timely interventions, are essential to improve patient outcomes [21].

Machine learning (ML) is a branch of artificial intelligence that focuses on training computers to learn from data collected and make improvements based on learned experience, and it is concerned with the problem of constructing computer programs that can automatically improve the accuracy of their output based on experience [22]. Recently, ML has been increasingly introduced into clinical practice, with applications including preclinical data processing, bedside diagnostic assistance, patient stratification, treatment decision-making, and early warning as part of primary and secondary prevention [8,23,24]. ML can improve clinical decision-making in a variety of ways by providing early warning, facilitating diagnosis, performing extensive screening, tailored treatment, and assessing patient response to treatment, and is increasingly being adopted from the well-established preclinical scene for a variety of fields and clinical applications [25,26,27]. In recent years, some researchers have applied ML techniques for mortality prediction in HF patients. For example, Negassa et al., developed an ensemble model for the prediction of 30-day mortality in patients with HF after discharge from hospital [28]. Adler et al., developed a new method to assess the risk of death in patients with HF using ML technique, namely MARKER-HF [29]. Jing proposed an ML model to accurately predict one-year all-cause mortality in a large number of HF patients, which can be used not only to stratify patients according to risk but also to effectively prioritize patients based on the predicted benefits of clinically relevant evidence-based interventions [30]. El-Rashidy also stated that after reviewing the past literature, no ML technique is completely superior to other methods in prediction and no developed system is commonly used for prediction due to low recognition rates [13]. Gasillas describes new machine learning algorithms for predicting mortality in COVID-19 patients [31]. Bi et al. presented the study focused on assessing the predictive performance of ML methods for in-hospital mortality in adult PCS patients [32]. José et al. introduced a new methodology to improve ICU monitoring systems through an age-based stratification approach, using XGBoost classifiers and SHAP technology to automatically identify thresholds for the most important clinical variables to monitor interpretable machine learning [33]. Therefore, accurate prediction of hospital mortality has remained a challenge to date.

Given that each ML method may outperform or have shortcomings in different situations, developing a model that integrates multiple ML methods to obtain better performance has become a new research approach. There are three main types of ensemble learning methods: Boosting, Bagging, and Stacking. Boosting updates the weights of the training data after each training iteration, and then classifies the output by weighted voting combinations [34]. Bagging involves training several base learners with different bootstrap samples, then consolidating them and voting on the result [35]. Stacking is a powerful ensemble technique that harnesses the predictions of multiple base learners as features to train new meta learners [36]. Stacking usually performs better than any single trained model. Li et al. [37] proposed a stacked Long Short-Term Memory (LSTM) network migration learning framework to improve the migrability of the traffic prediction model. Zhai & Chen [38] used a stacked ensemble model to predict and analyze the daily average concentration of PM2.5 in Beijing. Jia [39] proposed a stacked approach for ML to efficiently and rapidly construct 3D multi-type rock models using geological and geophysical datasets. Zhou et al. [40] investigated the stacking algorithm for cheating detection in large-scale assessments with consideration of class imbalance. Meharie et al. [41] presented a stacking ensemble algorithm to linearly and optimally estimate the cost of the highway construction project. Dong et al. [42] in order to improve the wind power forecasting, proposed an ensemble learning model based on stacking framework. In this paper, we deployed the stacking method to perform mortality prediction model construction for HF patients in ICU, which consists of Random Forest (RF), Support Vector Classifier (SVC), K-Nearest Neighbors (KNN), Light Gradient Boosting Machine (LGBM), Bootstrap aggregating (Bagging), and Adaptive Boosting (AdaBoost).

In this study, we used detailed clinical data from the MIMIC-III database, and focused on the collected data of ICU center patients suffering from heart diseases to predict the mortality of HF patients at 3 days, 30 days, 3 months and 1 year after admission to ICU using ML modeling techniques. This study focuses on proposing a stacking-based model to predict mortality in patients with HF. The base estimators can be adaptively selected and applied to the base layer of the stacking model. In addition, in the selection of key variables, it is expected that the constructed model can also successfully identify the important vital signs indicators of HF patients. We hope to improve the prediction of mortality in patients with heart disease by medical professionals, so that patients, their families and medical professionals are afforded with more informed judgments and make more appropriate prognostic preparations. The contributions of this study are summarized below.

We propose an accurate and medically intuitive framework for predicting mortality in the ICU based on a comprehensive list of key characteristics of patients with HF in the ICU. The model is based on ensemble learning theory and stacking methods, and constructs a heterogeneous ensemble learning model to improve the generalization and prediction performance of the model.For our model we adopted the most popular and most diverse classifiers in current literature, including six different ML techniques. The generated classifier lists are used to construct the proposed stacking models. The different meta classifiers were tested and the best performing estimators were selected. The predictive capabilities of our stacking model outperform the results of a single classifier and standard ensemble techniques, achieving encouraging accuracy and strong generalization performance.In addition, feature importance provides a specific score for each feature, and these scores indicate the impact of each feature on model performance. The importance score represents the degree to which each input variable adds value to the decision in the constructed model. We also obtained the clinical characteristics of patients with HF in the ICU that had the greatest impact on model prediction.

## 2. Materials and Methods

### 2.1. Patient Selection and Variable Selection

The MIMIC-III dataset was used in this study. MIMIC-III integrates comprehensive clinical data of patients admitted to the Beth Israel Deaconess Medical Center (BIDMC) in Boston, Massachusetts. The dataset collected de-identified data on 46,520 ICU admissions from 2001 to 2012 [43]. These data consisted of 26 tables containing aggregate information such as demographics, admission records, discharge summaries, ICD-9 diagnosis records, vital signs, laboratory measures, medications, clinical vital signs measurements, nursing staff observation records, radiology reports, and survival data. We completed the National Institutes of Health online course and passed the Protecting Human Research Participants examination and received data access privileges (Certificate No. 35628530).

For this study, in order to prevent possible information leakage and ensure a similar experimental setting compared to the related work, we used only the first ICU admission data for each patient. To emphasize the early predictive value, we used data from the first 24 h of patient admission as input to the predictive model and excluded patients younger than 16 years of age [12,21,26,44,45,46]. A patient cohort was selected based on the following exclusion criteria: The first ICU stay without all subsequent ICU stays, single ICU stay more than 24 h, patients’ first 24 h of ICU admission, and adult patients (age ≥ 16). Patients with discharge diagnosis as HF were screened out based on the International Classification of Diseases, 9th Revision codes (ICD-9) 398.91, 402.01, 402.11, 402.91, 404.01, 404.03, 404.11, 404.13, 404.91, 404.93, 428.xx [47,48,49,50]. After the first stage of data screening, a total of 7278 eligible patients were qualified.

The data for the predictive variables selected in this paper were obtained from two tables: admission table and chartevents table in the MIIMIC-III database. As a result of the study, we referred to the predictive variables that have been used by other researchers to predict HF mortality in patients [2,3,47,48,50] by measuring clinical symptoms 24 h before admission to the ICU, including Heart Rate, Respiratory Rate, Diastolic Blood Pressure, Systolic Blood Pressure, Temperature, Oxygen Saturation, Blood Urea Nitrogen, Creatinine, Mean Blood Pressure, Glucose, White Blood Cell, Red Blood Cell, Prothrombin Time, International Normalized Ratio, Platelets, GCS Eye, GCS Motor, GCS Verbal, and Patient’s Age and Gender.

In addition, for the treatment of missing values, the data preprocessing method of Guo et al., [21] was adopted in this study to perform a three-stage missing value treatment. Firstly, patients with more than 30% missing values were removed, thus 579 patients were removed. Secondly, variables with more than 40% missing values of the predictor variables were removed, so 4 variables were removed and 16 predictor variables were finally used in this study. Finally, statistics with missing values greater than 20% for these indicators were removed, and the remaining missing values were interpolated using menus. Finally, a total of 6699 patients were used in this study. Figure 1 shows the detailed process of data extraction.

### 2.2. Proposed Framework

Mortality rate is a major outcome in acute care: ICU mortality is the highest among hospital units (10% to 29%, depending on age and disease), and early identification of high-risk patients is key to improving outcomes [51]. Mortality prediction is one of the primary outcomes of hospitalization. In this paper, we use ICU admission records of HF patients in the MIMIC-III dataset to predict mortality in HF patients at different time points. We propose an ensemble learning model based on stacking. In the first layer of the stacking model we applied six ML methods, including RF, SVC, KNN, LGBM, Bagging and Adaboost to perform a first-level classifier on the dataset. Then, the fusion of these six classifier decisions was used to construct and optimize the stacked set of classifiers. This was followed by subjecting the datasets to second-level classifier using LGBM, and formed the final prediction. Figure 2 shows the Stacking ensemble based on a cross-validation of all feature subsets.

In this study, the most popular and diverse classifiers in related literature were applied in the mortality prediction model for patients with HF in the ICU. Six commonly used ML algorithms include: Random Forest (RF), Support Vector Classification (SVC), K-Nearest Neighbors (KNN), and Light Gradient Boosting Machine (LGBM), Bootstrap aggregating (Bagging), and Adaptive Boosting (AdaBoost). The values of the parameters for each model are shown in Table 1 [52].

Random Forest (RF) is an ensemble supervised ML algorithm. It uses decision trees as the basic classifier. RF generates many classifiers and combines their results by majority voting [53]. In the regression model, the output categories are numerical and the mean or average of the predicted outputs is used. The random forest algorithm is well suited to handle datasets with missing values. It also performs well on large datasets and can sort the features by importance. The advantage of using RF is that the algorithm provides higher accuracy compared to a single decision tree, it can handle datasets with a large number of predictive variables, and it can be used for variable selection [54].Support Vector Classifier (SVC) performs classification and regression analysis on linear and non-linear data. SVC aims to identify classes by creating decision hyperplanes in a non-linear manner in a higher eigenspace [55]. SVC is a robust tool to address data bias and variance and leads to accurate prediction of binary or multiclass classifications. In addition, SVC is robust to overfitting and has significant generalization capabilities [56].The K-Nearest Neighbors (KNN) algorithm does not require training. It is used to predict binary or sequential outputs. The data is divided into clusters and the number of nearest neighbors is specified by declaring the value of “K”, a constant. KNN is an algorithm [53] that stores all available instances and classifies new instances based on a similarity measure (e.g., distance function). Due to its simple implementation and excellent performance, it has been widely used in classification and regression prediction problems [57].Light Gradient Boosting Machine (LGBM) is an ensemble approach that combines predictions from multiple decision trees to make well-generalized final predictions. LGBM divides the consecutive eigenvalues into K intervals and selects the demarcation points from the K values. This process greatly accelerates the prediction speed and reduces storage space required without degrading the prediction accuracy [58,59]. LGBM is a gradient boosting decision tree learning algorithm that has been widely used for feature selection, classification and regression [60].The Bootstrap aggregating (Bagging) algorithm, also known as bagging algorithm, is an ensemble learning algorithm in the field of machine learning. It was originally proposed by Leo Breiman in 1994. Bagging algorithm can be combined with other classification and regression algorithms to improve its accuracy, stability, and avoid overfitting by reducing the variance of the results. Bagging is an ensemble method, i.e., a method of combining multiple predictors. It helps to avoid overfitting and variance reduction of the model to the data and has been used in a series of microarray studies [61,62]. We implemented Bagging using python’s sklearn library. We chose an ensemble of 500 DecisionTreeClassifier classifiers with a maximum sample set of 100 for each classifier, sampled each time using self-sampling, and trained all other hyperparameters by applying the sklearn default values.The self-adaptive nature of the Adaptive Boosting (AdaBoost) method is that the wrong samples of the previous classifier are used to train the next classifier, therefore, the AdaBoost method is sensitive to noisy data and anomalous data. It trains a basic classifier and assigns higher weights to the misclassified samples. After that, it is applied to the next process. The iterative process continues until the stopping condition is reached or the error rate becomes small enough [63,64]. We implemented AdaBoost using python’s sklearn library, choosing a maximum number of iterations of 50 for our hyperparameters and using the default values in sklearn for the rest of the training model.

### 2.3. Stacking Ensemble Technique

Stacking is an ensemble method for connecting multiple different types of classification models through a meta-classifier [65]. The basic concept is to combine multiple weak learners to obtain a model with stronger generalization ability [66]. Stacking is a novel ensemble framework in ensemble learning that uses meta-learners to fuse the results generated by each base learner [67]. Generally, base learners are called first-level learners and combinators are called second-level learners or meta learners. The basic principle of stacking is as follows. First, the first-level learner is trained using the initial training dataset. Then, the output of the first-level learner is used as the input feature of the meta learner. Finally, a new dataset is formed using the corresponding original labels as new labels to train the meta learner. If the first-level learner uses the same type of learning algorithm, then it is called homogeneous ensembles, otherwise it is called heterogeneous ensembles [42,68,69,70].

In the study, the first-layer prediction model of the stacking ensemble model was trained using a k-fold cross-validation method. The specific training process is as follows:The original dataset *S* is randomly divided into *K* sub-datasets {*S*_1_, *S*_2_, ⋯, *S_n_*}. Taking base learner 1 as an example, each sub-dataset *Si* (*i* = 1, 2, ⋯, *K*) is verified separately, and the remaining *K* − 1 sub-datasets are used as training sets to obtain *K* prediction results. Merge into set *D*_1_, which has the same length as *S*.Perform the same operation on other *n* − 1 base learners to obtain the set *D*_2_, *D*_3_, ⋯, *D_n_*. Combining the prediction results of *n* base learners, a new dataset *D* = {*D*_1_, *D*_2_, ⋯, *D_n_*} is obtained, which constitutes the input data of the second-layer meta-learner.The second-layer prediction model can detect and correct errors in the first-layer prediction model in time, and improve the accuracy of the prediction model.

As heterogeneous ensembles have better generalization performance and prediction accuracy, this study proposes a stacked ensemble classifier that can be divided into two stages. First, we use RF, SVC, KNN, LGBM, Bagging and Adaboost as the base classifiers in the first stage. The individual six classification models are trained using the complete training set; then, the probabilistic outputs obtained in the first stage are fed into the meta-classifier in the second stage, and then the meta-classifier is fitted based on the output meta-features of each classification model using the chosen ensemble techniques. The meta classifier can be trained on the predicted category labels or probabilities from the ensemble technique.

### 2.4. Synthetic Minority Oversampling Technique (SMOTE)

In ML, the problem is imbalanced when the class distribution is highly skewed. Imbalanced classification problems usually occur in many applications and pose obstacles to traditional learning algorithms [71]. In general, an imbalanced dataset may adversely affect the results of the model. Gold standard datasets are usually imbalanced, which will lead to a decrease in the predictive power of the model [66]. Model overfitting and underfitting are the most common problems encountered when evaluating performance. Overfitting occurs when a model shows high accuracy scores during training and low accuracy scores during validation. By adding more data to the training set and reducing the number of layers in the neural network, overfitting of models can be minimized. Underfitting occurs when the model fails to classify the data or make predictions during the training stage. SMOTE is a powerful solution to the classification imbalance problem and has delivered robust results in various domains [72]. The SMOTE algorithm adds synthetic data to a small number of classes to form a balanced dataset [71]. Class imbalance refers to the disproportionality between the classes of data used to train predictive models, a common problem that is not unique to medical data. Classification algorithms tend to favor the majority classes when the training data with negative results have significantly fewer observations than the classes with the majority observations, therefore, class imbalance problems can be resolved by manipulating the data, the algorithm, or both, to improve the predictive performance [6]. The core of the method is to perform random undersampling and oversampling for larger samples and smaller samples, respectively. In the MIMIC-III used in this study, only a few patients died during their ICU admission. Therefore, the SMOTE method, which uses synthetic minority sampling techniques to preprocess highly imbalanced data sets, was used in this study.

In our study, we applied the SMOTE techniques with different percentages for different cases. As a result, Table 2 several new training datasets were generated. Take the 3 days mortality dataset as an example, SMOTE with (3000%) increased the sample with class “died” from 213 instances to 6390 instances. This made an incremental increase in the minority class from 3.18% in the original dataset to 49.63% in the SMOTE with 3000% dataset.

### 2.5. Evaluation Criteria

The performance of the full classification is measured using different evaluation parameters, which consist of binary values (positive and negative). Two general evaluation measures, precision and recall, were used to evaluate the sentiment of tweets based on positive and negative polarity, including Accuracy and F-score for micro averaging purposes. Four functional accuracy measures were taken into account based on the outcomes of the confusion matrix named true positive (TP), true negative (TN), false positive (FP), and false negative (FN). In this study, five evaluation tools were selected as performance indicators. We used different indicators in order to have a better understanding of the results. The evaluation parameters used to measure the performance of our proposed system are listed below: we applied Precision, Recall, F-score, and Accuracy, which are widely used in the research field, to evaluate the results of our study, defined as follows:(1)$$Precision=PPV=\frac{TP}{TP+FP}$$
(2)$$Recall=TPR=\frac{TP}{TP+FN}$$
(3)$$F-score=\frac{2*Precision*Recall\;}{Precision+Recall\;}$$
(4)$$Accuracy=\frac{TP+TN}{TP+FP+TN+FN}$$

In addition, this study adopts the area under the receiver operating characteristic (AUROC) to measure the predictive performance of the model. AUROC represents the generalization ability of deep learning, which is the area enclosed by the ROC curve (the curve composed of true positive rate and false positive rate). The ROC curve is widely used in dichotomous problems to evaluate the reliability of the classifier. The FPR value on the horizontal coordinates and the TPR value on the vertical coordinates yield a curve called the ROC curve, and the metric considered is the AUC of the curve, called AUROC, with AUC values ranging from 0 to 1; the larger value the better [73]. The ROC curve is not affected by the imbalance in the distribution of the dataset, allowing a more objective evaluation of the performance in the case of unbalanced learning. A completely threshold-independent AUC can be computed to reveal the performance of the classification algorithm. A larger value of AUC implies that the classification algorithm exhibits stronger and more stable predictive performance [58].

## 3. Results

### 3.1. Baseline Characteristics

The study was preprocessed with data from the MIMIC-III database and ultimately used the ICU admission records of 6699 HF patients. Table 3 provides demographic information on HF patients. The mean age of the patients in this study was 70.3 years, 55% of whom were male, and the mean number of days hospitalized and the mean number of days in the ICU was 5.8 days. In addition, more than 84% of the patients were hospitalized for medical emergencies. 37.9% of patients were admitted to the Medical ICU and 27.7% were admitted to the Coronary Care Unit. Furthermore, for patient insurance, over 70% of patients were on Medicare. Table 3 also presents the mean of the 20 variables used in this study.

### 3.2. Mortality Prediction Results of Different Models

In this study, a 10-fold cross-validated training and testing was used, with 80% of the data used for training the model and 20% for testing the model, followed by extensive statistical analysis to evaluate performance. In this study, data collected within 24 h of admission to the ICU were used to predict the mortality of patients 3 days, 30 days, 3 months and 1 year after admission. The most commonly used metric to assess the performance of diagnostic tools is AUROC, which graphically presents the true positive rate and the false positive rate. Table 4 lists the six different ML methods used in the first phase of this study, and the addition of the Stacking technique in the second phase involved seven techniques to generate the AUROC of HF patient mortality prediction tasks over four different time periods. This study determined the highest AUROC for predicting 3 days mortality in patients with HF, i.e., death within 3 days could be accurately predicted within 24 h of patient admission. Figure 3 and Figure 4 indicate that the data collected from ICU admissions can be used to predict mortality within 3 days, which is a better prediction outcome; Figure 3 also clearly shows that all four models have high AUROC after adding stacking technique in the second stage, which means they can distinguish mortality from non-mortality cases well. Therefore, prediction models constructed in this study, can predict the life and health status of patients more accurately.

In addition to the above AUROC, this study also compares Precision, Recall, F-score, and Accuracy, as shown in Table 5, when evaluating the performance of different attribute sets used in the classification algorithm. The best precision and recall was shown in a run where the 3 days mortality was used to select the best subset of attributes. In addition, it can be observed from Figure 5 that the addition of Stacking technique in the second stage results in a higher accuracy, with the highest value of 0.9525 for predicting mortality within 3 days. The highest value was 0.9525 for predicting mortality within 3 days. It also demonstrated that this study could achieve very good results in predicting mortality in HF patients using data from 24 h before the patients were admitted to ICU.

### 3.3. Interpretation of Variable Importance

Feature importance is the main contribution of each feature to improve the predictive power of the whole model. It provides an intuitive view of the importance of features to see which features have a greater impact on the final model, but it is not possible to determine how the features relate to the final prediction. Figure 6 lists the different explanatory variables according to their contribution to each model. We can observe that the clinical characteristics of Platelets, Glucose, Blood Urea Nitrogen, Age, Heart Rate, Systolic Blood Pressure, and Diastolic Blood Pressure are important factors influencing the prediction of HF, and previous studies of HF Similar results have been found in previous studies of HF [74,75,76]. However, the effects of GCS eye, GCS motor, GCS verbal, Red Blood Cell, International Normalized Ratio and Gender were not as significant.

In this study, we used several variables for predicting mortality in patients with HF in the ICU with reference to previous related studies. These characteristics result in different contributions of the parameters in the input layer in the construction of LightGBM. The LightGBM model was originally proposed by Microsoft^®^ [77]. It is a decision tree-based algorithm that divides the parameters in the input layer into different parts to construct a mapping relationship between inputs and outputs. More specifically, the LightGBM model uses a gradient boosted decision tree (GBDT) algorithm to achieve more accurate predictions. In the LightGBM model, many hyperparameters are used, but since hyperparameters can significantly affect model performance, many of them can be fine tuned for different applications to improve model performance. Therefore, hyperparameter tuning is an essential process in the construction of ML models. Figure 6 shows the contribution of the first 20 variables in the input layer at each station. Parameter contribution represents the ratio of the number of times a parameter is used for tree splitting to the total number of times all parameters are used for tree splitting. A larger parameter contribution value means that it has been used more times to split nodes, indicating a more significant parameter importance. For a deeper understanding of the LightGBM algorithm, please refer to Guolin Ke et al. [77], which describes the principles and applications of the LightGBM algorithm in detail. In addition, more information about how the LightGBM method calculates the importance of input variables can be found in the related research note [78,79].

Table 6 lists the top ten most important clinical features that contributed most to the model at four different prediction times. The contributions are ranked from most important to least important. We can find that, in addition to considering an HF patient’s Platelets, Glucose, and Blood Urea Nitrogen, Systolic Blood Pressure and Diastolic Blood Pressure are also important variables in predicting 3 days, 30 days, and 3 months mortality of HF patients in the ICU. Heart Rate is an important variable to be considered when predicting 1 year mortality in patients with HF in ICU.

## 4. Discussion

The feasibility of the ML technique for mortality prediction in HF patients has been previously demonstrated. Negassa et al., developed an ensemble model for 30-day post-discharge mortality and used discrimination, range of prediction, Brier index and explained variance as metrics to evaluate model performance, the discrimination achieved by the ensemble model was higher 0.83 [28].Adler et al. developed a new approach to risk assessment using ML techniques MARKER-HF algorithm, which naturally captures correlations in the covariates’ multi-dimensional space predicted by a separate U.S. healthcare system and a large European registry mortality with AUCs of 0.84 and 0.81, respectively [29]. Jing proposed an ML model to accurately predict one-year all-cause mortality in a large number of HF patients, with the nonlinear XGBoost model (AUC: 0.77) achieving the best prediction [30]. Austin et al. developed ML algorithms to predict mortality in 12,608 patients with acute HF, the baseline logistic regression model with the AUC of 0.794 [80]. Luo et al., constructed a risk stratification tool using the extreme gradient boosting algorithm to correlate patient clinical characteristics with in-hospital mortality, and this new ML model outperformed traditional risk prediction methods with an AUC of 0.831 [81]. The aim of this study is to predict the mortality of ICU patients by ML model from structural vital signs data collected during the ICU stay of HF patients.

Through in-depth analysis of the experimental results, the following conclusions were drawn from this study: (1) The proposed stacking ensemble model can make full use of the observation ability of different prediction models to the data space and structure, so that different models can learn from each other, so that it is possible to obtain the best prediction. (2) The data collected during the first 24 h after the admission of HF patients to the ICU were used for modeling analysis, and the ML stacking method was effective and rapid in constructing predictive models. The results indicated that our proposed stacking method has good performance in predicting mortality in HF patients with Accuracy (95.25%) and AUROC (82.55%). This also validates the findings of an analysis of the relationship between classifier diversity and the quality of stacking and concluded that diversity may be considered as the selection criteria for building the ensemble classifier [13,82]. (3) This study was able to accurately predict the mortality of patients after admission to the ICU; as the prediction time was shortened, the prediction performance became better. (4) Random Forest, LGBM and Bagging ML techniques have been rapidly developed in recent times to accurately predict outcomes in the medical field. (5) This study also found that Platelets, Glucose, and Blood Urea Nitrogen were the clinical characteristics that had the greatest impact on model prediction and were important indicators to be considered in the selection of important variables, which is in line with previous studies on HF [74,76].

However, there are some limitations to this study. First, this study is not a prospective study, but a retrospective study, and inherent biases are inevitable in retrospective studies. Since most of the dynamic data obtained from ICU is used to train the classifier specifically, there may be problems in non-ICU settings; i.e., less real-time data may negatively affect the prediction results and increase the possibility of incorrect judgments and decisions. This is a common problem in the practical implementation of ML methods using dynamic data from EHR [83]. Second, in this study, in order to consider the convenience and completeness of data collection, only ICU databases with the easiest access to complete dynamic patient information were considered. The MIMIC-III data used in this study were obtained from the Beth Israel Deaconess Single Medical Center in Boston, Massachusetts, USA. We trained our ML model using ten-fold cross-validation and tested the final model using an independent patient cohort as an external validation of model generalizability. We evaluated the performance of all models on the data set using Scikit-learn v0.23.1 [52]. However, the broader application of our ML model requires further validation in different statistical populations. More comprehensive results and validation may be obtained for patients in other types of medical institutions such as large hospitals or small clinics, or in medical institutions of other scales. Third, the data used in this study were limited to the data collected during the first ICU stay, excluding the records and reports of patients readmitted to the hospital. The collection of data from multiple ICU hospitalizations may provide a comprehensive assessment of time series issues, or may provide more different levels of analysis to patients, healthcare professionals, and patients’ families for future evaluation and prognosis. Fourth, the data extracted from the MIMIC database were distributed over multiple years (2001–2012), during which the treatment of HF changed significantly, which may weaken the application of our model.

## 5. Conclusions

The world is facing the challenge of a COVID-19 pandemic, where patients need regular monitoring of vital signs and medical warning scores to enable early identification and treatment of their disease. The burden on the healthcare system is enormous, and the interventions used and their duration may seriously affect the prognosis of patients [11]. Despite the recent development of various severity scores and ML models for early mortality prediction, such predictions remain challenging. Compared to existing solutions, this study provides a complement to current clinical decision-making methods by proposing a new stacked ensemble approach to predict mortality in patients with HF in the ICU. The base estimators can be adaptively selected and applied to the base layer of the stacking model. Our stacking model is significantly better than the traditional ML approach in mortality prediction, and can successfully screen out the important clinical features of HF patients. It can empower health care professionals to better predict mortality in HF patients, and provide patients, their families and medical professionals with more information to determine the status of patients and make more appropriate prognosis. For follow-up studies, this study suggests the following recommendations for future studies:Compared to structured data that have been used for clinical outcome prediction, the information available in diagnostic records and test reports in unstructured data is still underutilized by medical research. These diagnostic data are important references for clinical decision-making because they record multifaceted information about the patient’s visit, such as the focus of care, preliminary medical assessment, and the generation of different recommendations for the final diagnosis. Future research suggests that structured data and unstructured data can be integrated for more detailed classification and study [84].The MIMIC-III data is relatively rich and complete, and this study only modeled and predicted the mortality of patients; subsequent studies can be conducted to evaluate the readmission, length of stay, medication use, and complications of patients with reference to the framework of this study. This type of study can be made more objective and complete if it can be extended to conduct more comprehensive evaluation and analysis.The evolution of variables over time can be collected from patient EHR data in an attempt to obtain better predictive effects. In terms of research methods, future research can attempt different ML methods as well as deep learning methods that have recently been applied to solve time-series data more effectively. For example, long short-term memory, recurrent neural network, CNN models [85,86] are common deep learning models.With the popularity and increasing prevalence of AI, telemedicine and robotics, which have emerged in response to the recent COVID-19, imaging AI and speech AI can be incorporated. Combining existing clinical data, diagnostic reports, medical image images, etc., can improve medical culture and quality of care, which will be an important issue in the future field of smart medicine [87,88].

## Figures and Tables

**Figure 1 jcm-11-06460-f001:**
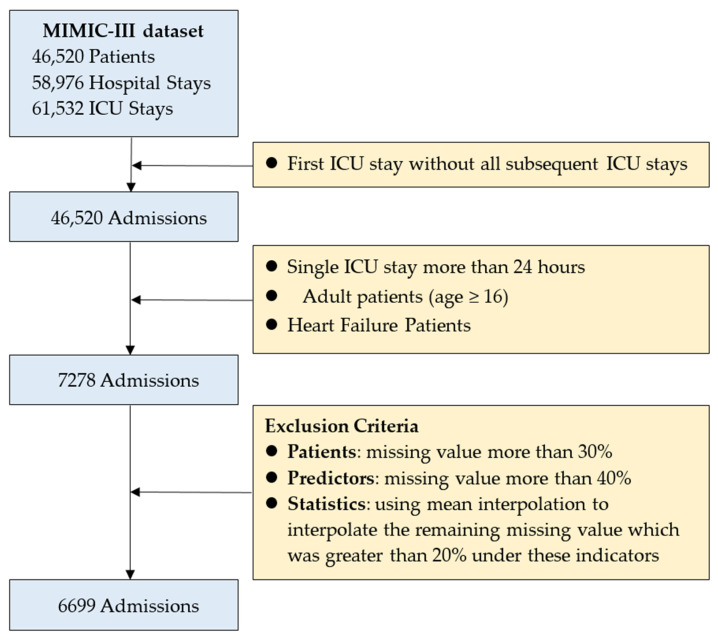
The detailed process of data extraction.

**Figure 2 jcm-11-06460-f002:**
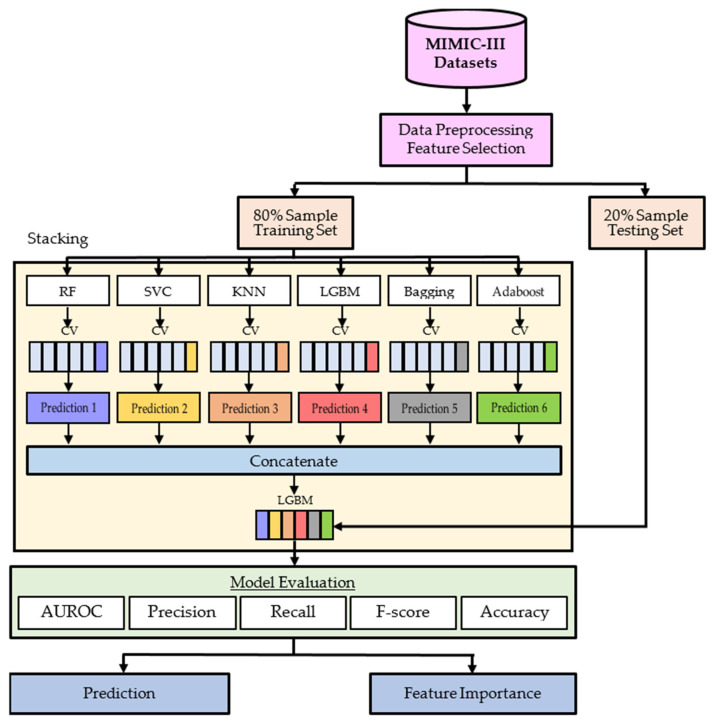
Stacking ensemble based on a cross-validation of all feature subsets.

**Figure 3 jcm-11-06460-f003:**
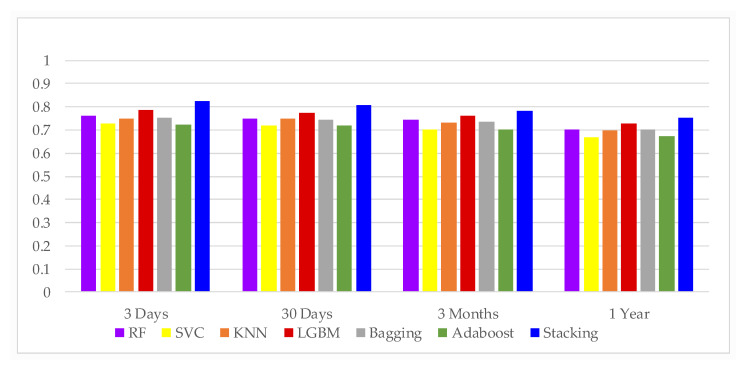
The AUROCs of different classifiers.

**Figure 4 jcm-11-06460-f004:**
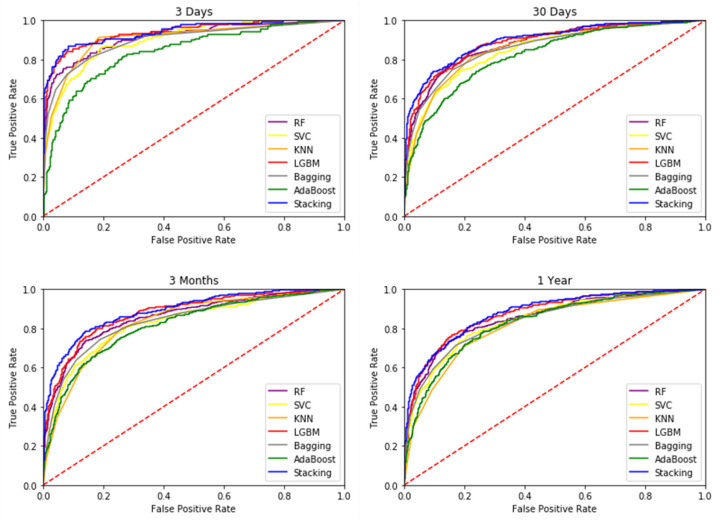
The ROC curves of all base models.

**Figure 5 jcm-11-06460-f005:**
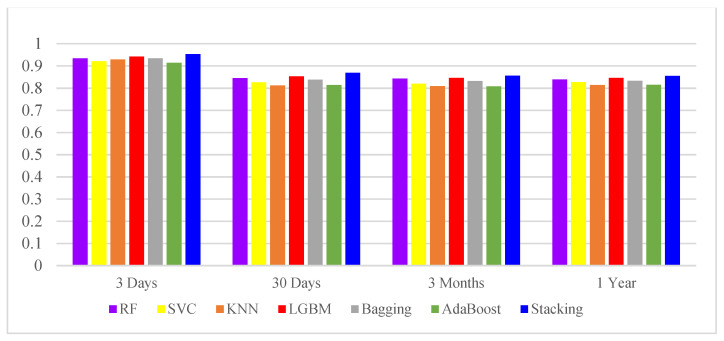
The Accuracy of different classifiers.

**Figure 6 jcm-11-06460-f006:**
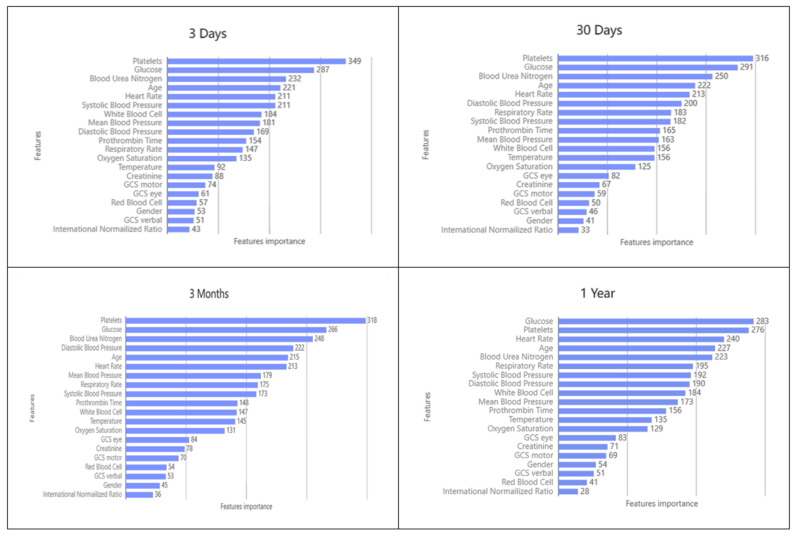
The contribution of the first 20 important variables by using LGBM.

**Table 1 jcm-11-06460-t001:** The parameters of the six candidate models.

Method	Parameters
RF	*max*_*depth* = *20*, *min*_*samples*_*split* = *0.001*, *n*_*estimators* = *20*
SVC	*C* = *1.0*, *kernel* = ‘*rbf*’, *degree* = *3*, *gamma* = ‘*auto*’, *coef0* = *0.0*, *shrinking* = *True*, *probability* = *False*, *tol* = *0.001*, *cache*_*size* = *200*, *class*_*weight* = *None*, *verbose* = *False*, *max*_*iter* = −*1*
KNN	*n*_*neighbors* = *5*, *weights* = ‘*uniform*’, *algorithm* = ‘*auto*’, *leaf*_*size* = *30*, *p* = *2*, *metric* = ‘*minkowski*’
LGBM	*boosting*_*type* = ‘*gbdt*’, *num*_*leaves* = *31*, *max*_*depth* = −*1*, *learning*_*rate* = *0.1*, *n*_*estimators* = *100*, *subsample*_*for*_*bin* = *200*,*000*
	*min*_*child*_*samples* = *20*, *subsample* = *1.0*, *subsample*_*freq* = *0*, *colsample*_*bytree* = *1.0*, *reg*_*alpha* = *0.0*, *reg*_*lambda* = *0.0*
	*n*_*jobs* = *−1*, *importance*_*type* = ‘*split*’
Bagging	*base*_*estimator* = *None*, *n*_*estimators* = *10*, *max*_*samples* = *1.0*, *max*_*features* = *1.0*, *bootstrap* = *True*, *bootstrap*_*features* = *False*, *oob*_*score* = *False*, *warm*_*start* = *False*, *n*_*jobs* = *None*, *random*_*state* = *None*, *verbose* = *0*
Adaboost	*base*_*estimator* = *DecistionTreeClassifer*, *random*_*state* = *1*, *n*_*estimators* = *50*, *learning*_*rate* = *1.0*, *algorithm* = ‘*SAMME.R*’

RF, Random Forest; SVC, Support Vector Classification; KNN, K-Nearest Neighbors; LGBM, Light Gradient Boosting Machine; Bagging, Bootstrap aggregating; Adaboost, Adaptive Boosting.

**Table 2 jcm-11-06460-t002:** Number of instances increased by SMOTE technique.

	Number of Survive	Number of Death	Percentage of SMOTE Increase	Class “Survived”	Class “Died”
3 Days	6486	213	3000%	6486	6390
30 Days	5822	877	600%	5822	5262
3 Months	5765	934	600%	5765	5604
1 Year	5754	945	600%	5754	5670

**Table 3 jcm-11-06460-t003:** Features involved in the model.

	Overall	Alive at ICU	Dead at ICU
General			
Number	6699 (100%)	5754 (85.89%)	945 (14.11%)
Age (Q1–Q3)	70.31 ± 13.04	69.88 ± 13.03	72.92 ± 12.74
Gender (male)	3694 (55.14%)	3185 (55.35%)	509 (53.86%)
Outcomes			
Hospital LOS (days) (Q1–Q3)	13.04 (5.99–16.00)	12.78 (6.06–15.77)	14.60 (5.27–19.09)
ICU LOS (days) (Q1–Q3)	5.79 (1.93–6.23)	5.40 (1.88–5.77)	8.17 (2.38–10.38)
Admission Type			
ELECTIVE	766 (11.43%)	721 (12.53%)	45 (4.76%)
EMERGENCY	5668 (84.61%)	4807 (83.54%)	861 (91.11%)
URGENT	265 (3.96%)	226 (3.93%)	39 (4.13%)
Care Unit Type			
CCU	1852 (27.65%)	1633 (28.38%)	219 (23.17%)
CSRU	1319 (19.69)	1240 (21.55%)	79 (8.36%)
MICU	2537 (37.87%)	2049 (35.61%)	488 (51.64%)
SICU	635 (9.48%)	526 (9.14%)	109 (11.34%)
TSICU	356 (5.31%)	306 (5.32%)	50 (5.29%)
Insurance			
Government	96 (1.43%)	88 (1.53%)	8 (0.85%)
Medicaid	388 (5.79%)	352 (6.12%)	36 (3.81%)
Medicare	4748 (70.88%)	4014 (69.76%)	734 (77.67%)
Private	1439 (21.48%)	1277 (22.19%)	162 (17.14%)
Self Pay	28 (0.42%)	23 (0.40%)	5 (0.53%)
Variable value			
Heart Rate	85.60 ± 15.49	85.03 ± 15.15	89.04 ± 16.98
Respiratory Rate	19.51 ± 4.18	19.31 ± 4.01	20.74 ± 4.94
Diastolic Blood Pressure	57.10 ± 12.15	57.57 ± 12.15	54.35 ± 11.73
Systolic Blood Pressure	115.01 ± 19.11	115.70 ± 19.04	110.98 ± 19.01
Temperature	98.18 ± 1.42	98.21 ± 1.38	98.02 ± 1.60
Oxygen Saturation	97.00 ± 2.24	97.09 ± 2.02	96.51 ± 3.25
Fractional Inspired Oxygen	15.55 ± 26.57	16.12 ± 26.69	12.73 ± 25.81
Blood Urea Nitrogen	33.53 ± 24.12	31.73 ± 22.70	44.50 ± 29.12
Creatinine	1.76 ± 2.06	1.72 ± 2.13	1.99 ± 1.57
Mean Blood Pressure	76.97 ± 12.45	76.98 ± 11.50	76.93 ± 17.14
Glucose	146.11 ± 46.21	144.65 ± 45.09	154.76 ± 51.53
White Blood Cell	12.63 ± 11.94	12.21 ± 7.27	15.15 ± 26.12
Red Blood Cell	3.54 ± 0.52	3.54 ± 0.52	3.50 ± 0.55
Prothrombin Time	16.61 ± 6.74	16.43 ± 6.63	17.66 ± 7.24
International Normalized Ratio	1.65 ± 1.00	1.61 ± 0.94	1.87 ± 1.28
Platelets	216.09 ± 95.72	217.45 ± 93.55	207.80 ± 107.65
GCS eye	3.32 ± 0.86	3.38 ± 0.80	2.90 ± 1.05
GCS motor	5.33 ± 1.11	5.41 ± 1.01	4.88 ± 1.49
GCS verbal	3.44 ± 1.67	3.54 ± 1.64	2.80 ± 1.74

MICU Denotes Medical ICU; SICU Denotes Surgical ICU; CCU Denotes Coronary Care Unit; CSRU Denotes Cardiac Surgery Recovery Unit; TSICU Denotes Trauma Surgical ICU.

**Table 4 jcm-11-06460-t004:** The AUROCs of different classifiers.

	RF	SVC	KNN	LGBM	Bagging	Adaboost	Stacking
3 Days	0.7598 ± 0.0092	0.7249 ± 0.0176	0.7490 ± 0.0220	0.7868 ± 0.0084	0.7534 ± 0.0151	0.7230 ± 0.0076	0.8255 ± 0.0201
30 Days	0.7472 ± 0.0120	0.7179 ± 0.0086	0.7476 ± 0.0049	0.7724 ± 0.0086	0.7442 ± 0.0104	0.7168 ± 0.0078	0.8052 ± 0.0049
3 Months	0.7433 ± 0.0121	0.7002 ± 0.0145	0.7313 ± 0.0110	0.7596 ± 0.0078	0.7338 ± 0.0120	0.7005 ± 0.0121	0.7830 ± 0.0155
1 Year	0.6998 ± 0.0097	0.6671 ± 0.0081	0.6958 ± 0.0125	0.7269 ± 0.0062	0.7014 ± 0.0106	0.6706 ± 0.0048	0.7532 ± 0.0064

RF, Random Forest; SVC, Support Vector Classification; KNN, K-Nearest Neighbors; LGBM, Light Gradient Boosting Machine; Bagging, Bootstrap aggregating; Adaboost, Adaptive Boosting.

**Table 5 jcm-11-06460-t005:** Diagnostic Precision, Sensitivity, F-Score, and Accuracy of different classifiers.

	Method	Precision	Recall	F-Score	Accuracy
3 Days	RF	0.9167 ± 0.0328	0.3429 ± 0.0175	0.4989 ± 0.0210	0.9343 ± 0.0097
	SVC	0.8991 ± 0.0382	0.1920 ± 0.0352	0.3105 ± 0.0487	0.9208 ± 0.0114
	KNN	0.6635 ± 0.0207	0.5198 ± 0.0415	0.5824 ± 0.0328	0.9288 ± 0.0125
	LGBM	0.8763 ± 0.0532	0.5821 ± 0.0163	0.6989 ± 0.0223	0.9425 ± 0.0050
	Bagging	0.8176 ± 0.0616	0.3959 ± 0.0309	0.5322 ± 0.0324	0.9343 ± 0.0055
	AdaBoost	0.5937 ± 0.0633	0.3082 ± 0.0146	0.4044 ± 0.0202	0.9139 ± 0.0078
	Stacking	0.8030 ± 0.0108	0.6682 ± 0.0402	0.7286 ± 0.0223	0.9525 ± 0.0081
30 Days	RF	0.7857 ± 0.0217	0.5455 ± 0.0254	0.6435 ± 0.0201	0.8457 ± 0.0056
	SVC	0.7389 ± 0.0269	0.4962 ± 0.0173	0.5934 ± 0.0161	0.8262 ± 0.0050
	KNN	0.6399 ± 0.0242	0.6145 ± 0.0181	0.6264 ± 0.0105	0.8126 ± 0.0060
	LGBM	0.7704 ± 0.0104	0.6069 ± 0.0175	0.6789 ± 0.0144	0.8533 ± 0.0042
	Bagging	0.7526 ± 0.0246	0.5508 ± 0.0248	0.6355 ± 0.0171	0.8387 ± 0.0037
	AdaBoost	0.6827 ± 0.0380	0.5170 ± 0.0249	0.5871 ± 0.0136	0.8142 ± 0.0057
	Stacking	0.7831 ± 0.0171	0.6747 ± 0.0125	0.7247 ± 0.0086	0.8690 ± 0.0032
3 Months	RF	0.7878 ± 0.0102	0.5372 ± 0.0262	0.6385 ± 0.0200	0.8429 ± 0.0052
	SVC	0.7577 ± 0.0207	0.4505 ± 0.0284	0.5647 ± 0.0263	0.8206 ± 0.0085
	KNN	0.6507 ± 0.0149	0.5692 ± 0.0201	0.6072 ± 0.0176	0.8095 ± 0.0074
	LGBM	0.7712 ± 0.0118	0.5792 ± 0.0188	0.6613 ± 0.0119	0.8466 ± 0.0037
	Bagging	0.7476 ± 0.0226	0.5304 ± 0.0270	0.6200 ± 0.0201	0.8320 ± 0.0055
	AdaBoost	0.6844 ± 0.0057	0.4780 ± 0.0264	0.5625 ± 0.0195	0.8079 ± 0.0074
	Stacking	0.7720 ± 0.0114	0.6311 ± 0.0350	0.6939 ± 0.0221	0.8564 ± 0.0059
1 Year	RF	0.7600 ± 0.0175	0.4412 ± 0.0223	0.5577 ± 0.0156	0.8397 ± 0.0076
	SVC	0.7490 ± 0.0383	0.3722 ± 0.0221	0.4961 ± 0.0154	0.8267 ± 0.0103
	KNN	0.6252 ± 0.0196	0.4767 ± 0.0209	0.5408 ± 0.0198	0.8144 ± 0.0108
	LGBM	0.7405 ± 0.0231	0.5069 ± 0.0071	0.6017 ± 0.0107	0.8460 ± 0.0092
	Bagging	0.7114 ± 0.0172	0.4583 ± 0.0268	0.5569 ± 0.0192	0.8332 ± 0.0044
	AdaBoost	0.6558 ± 0.0149	0.4046 ± 0.0136	0.5001 ± 0.0090	0.8147 ± 0.0058
	Stacking	0.7428 ± 0.0229	0.5651 ± 0.0152	0.6414 ± 0.0093	0.8551 ± 0.0085

RF, Random Forest; SVC, Support Vector Classification; KNN, K-Nearest Neighbors; LGBM, Light Gradient Boosting Machine; Bagging, Bootstrap aggregating; Adaboost, Adaptive Boosting.

**Table 6 jcm-11-06460-t006:** The selected ten important variables by using LGBM.

Variable Importance	3 Days	30 Days	3 Months	1 Year
1	Platelets	Platelets	Platelets	Glucose
2	Glucose	Glucose	Glucose	Platelets
3	Blood Urea Nitrogen	Blood Urea Nitrogen	Blood Urea Nitrogen	Heart Rate
4	Age	Age	Diastolic Blood Pressure	Age
5	Systolic Blood Pressure	Heart Rate	Age	Blood Urea Nitrogen
6	Heart Rate	Diastolic Blood Pressure	Heart Rate	Respiratory Rate
7	White Blood Cell	Respiratory Rate	Mean Blood Pressure	Systolic Blood Pressure
8	Mean Blood Pressure	Systolic Blood Pressure	Respiratory Rate	Diastolic Blood Pressure
9	Diastolic Blood Pressure	Prothrombin Time	Systolic Blood Pressure	White Blood Cell
10	Prothrombin Time	Mean Blood Pressure	Prothrombin Time	Mean Blood Pressure

## Data Availability

Not applicable.

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
