# Peer review of "Applying an Improved Stacking Ensemble Model to Predict the Mortality of ICU Patients with Heart Failure"

_jcm, 2022, doi:10.3390/jcm11216460_

Round 1
Reviewer 1 Report
Write the complete Algorithm for the stacking ensemble model to predict the mortality of ICU patients with heart failure
Compare your results with recent research works
Add more recent, relevant literature
Author Response
We would like to thank you for your valuable recommendations that make the paper more complete and readable. We have made the following adjustments based on your suggestions.

Reviewer 2 Report
The work done in this paper seems very interesting to me, however the length of the text sometimes makes it difficult to read and makes the reader get lost. I would like both the introduction and the conclusions of the study to be simplified and shortened by the authors to encourage the extraction of more specific ideas from their work.
Author Response

(The authors gave the same response as above.)

Reviewer 3 Report
I would like to thank the authors for this work. The methodology is largely clear, and the manuscript is well-written as well. However, I have some points to consider in the next version, please.
(1)
The motivation behind the study needs further clarification. The literature already includes a plethora of studies that made use of the MIMIC dataset to develop similar predictive models. It is not entirely clear how the present study would fit within the existing literature.
(2)
I suggest reviewing the list of contributions mentioned in the introduction. The contributions claimed mostly describe the methodology, rather than a particular contribution. Specifically, the second and third points can be hardly described as part of the contributions.
(3)
I commend the authors on the effort put into the related work. However, the related work should also touch on recent topics, such as explainability of predictions. That said, part of the related work should refer to recent studies applying the MIMIC data in this regard. For example:
https://doi.org/10.3390/s21217125
https://doi.org/10.1109/ICHI48887.2020.9374393
(4)
Please elaborate further on the intuition behind the choice of stacking-based Machine Learning (ML). The reader would wonder why the authors preferred that approach over other state-of-the-art approaches, such as Deep Learning for example.
(5)
Please elaborate further on how SMOTE was applied. Please clarify how the synthetic samples were used during the training process. It is not entirely clear how the dataset was split while applying SMOTE.
(6)
A couple of references should be cited, please.
Chawla, N. V., Bowyer, K. W., Hall, L. O., & Kegelmeyer, W. P. (2002). SMOTE: synthetic minority over-sampling technique. Journal of Artificial Intelligence Research, 16, 321-357.
Pedregosa, F., Varoquaux, G., Gramfort, A., Michel, V., Thirion, B., Grisel, O., ... & Duchesnay, E. (2011). Scikit-learn: Machine learning in Python. Journal of Machine Learning Research, 12, 2825-2830.
Author Response

(The authors gave the same response as above.)

Round 2
Reviewer 2 Report
Good work
Reviewer 3 Report
Thanks for accommodating the feedback.
Please just revise the list of references. For conference papers, the publisher info should be specified as ‘In Proceedings of …’
In general, please follow the journal template in this regard.